# 5p and 3p Strands of miR-34 Family Members Have Differential Effects in Cell Proliferation, Migration, and Invasion in Cervical Cancer Cells

**DOI:** 10.3390/ijms20030545

**Published:** 2019-01-28

**Authors:** Sergio Córdova-Rivas, Ixamail Fraire-Soto, Andrea Mercado-Casas Torres, Luis Steven Servín-González, Angelica Judith Granados-López, Yamilé López-Hernández, Claudia Araceli Reyes-Estrada, Rosalinda Gutiérrez-Hernández, Julio Enrique Castañeda-Delgado, Leticia Ramírez-Hernández, José Antonio Varela-Silva, Jesús Adrián López

**Affiliations:** 1Laboratorio de microRNAs y Cáncer, Unidad Académica de Ciencias Biológicas, Universidad Autónoma de Zacatecas, Av. Preparatoria S/N, Zacatecas 98066, México; cordova092@hotmail.com (S.C.-R.); ixamail13@gmail.com (I.F.-S.); andrea.casas05@gmail.com (A.M.-C.T.); agranadosjudith@gmail.com (A.J.G.-L.); ylopezher@conacyt.mx (Y.L.-H.); antonio_varela@outlook.com (J.A.V.-S.); 2School of Life Sciences, Gibbert Hill Campus, University of Warwick, Coventry CV47AL, UK; L.Servin-Gonzalez@warwick.ac.uk; 3Laboratorio de Metabolómica de la Unidad Académica de Ciencias Biológicas, CONACyT, Universidad Autónoma de Zacatecas, Av. Preparatoria S/N, Zacatecas 98066, México; 4Laboratorio de Patología e Inmunohistoquímica de la Unidad Académica de Medicina Humana de la Universidad Autónoma de Zacatecas, Campus Siglo XXI, Kilómetro 6, Ejido la Escondida, Zacatecas CP 98160, Mexico; c_reyes13@yahoo.com.mx; 5Laboratorio de Etnofarmacología Nutrición de la Unidad Académica de Enfermería de la Universidad Autónoma de Zacatecas, Campus Siglo XXI, Kilómetro 6, Ejido la Escondida, Zacatecas CP 98160, Mexico; rosalindagh@hotmail.com; 6Catedrático-CONACYT, Unidad de Investigación Biomédica de Zacatecas, Instituto Mexicano del Seguro Social, Zacatecas CP 98000, Mexico; jecastanedade@conacyt.mx; 7Unidad Académica de Matemáticas de la Universidad Autónoma de Zacatecas Av. Preparatoria S/N, Zacatecas 98066, México; leticiaadrianaramirez@hotmail.com

**Keywords:** miR-34 family, guide strand, passenger strand, cell processes

## Abstract

The micro RNA (miR)-34 family is composed of 5p and 3p strands of miR-34a, miR-34b, and miR-34c. The 5p strand’s expression and function is studied in cervical cancer. The 3p strand’s function and regulation remain to be elucidated. To study the function of the passenger strands of miR-34 family members, we overexpressed 5p and 3p strands using a synthetic miRNA in cervical cell lines. Cell proliferation was evaluated using crystal violet. Migration and invasion were tested using transwell assays, Western blot, and zymography. Possible specific targets and cell signaling were investigated for each strand. We found that miR-34a-5p inhibited proliferation, migration, and cell invasion accompanied by matrix metalloproteinase 9 (MMP9) activity and microtubule-associated protein 2 (MAP2) protein reduction. We also found that miR-34b-5p and miR-34c-5p inhibit proliferation and migration, but not invasion. In contrast, miR-34c-5p inhibits MMP9 activity and MAP2 protein, while miR-34b-5p has no effect on these genes. Furthermore, miR-34a-3p and miR-34b-3p inhibit proliferation and migration, but not invasion, despite the later reducing MMP2 activity, while miR-34c-3p inhibit proliferation, migration, and cell invasion accompanied by MMP9 activity and MAP2 protein inhibition. The difference in cellular processes, MMP2 and MMP9 activity, and MAP2 protein inhibition by miR-34 family members suggests the participation of other regulated genes. This study provides insights into the roles of passenger strands (strand*) of the miR-34 family in cervical cancer.

## 1. Introduction

Cervical cancer is the fourth most common cancer in women worldwide, with an estimated global incidence of 570,000 new cases and over 311,000 deaths per year [1]. High-risk human papillomavirus (HPV) infection is associated with the development of cervical, penile, and anal cancers. Although high-risk HPV expression appears necessary to immortalize and stimulate proliferation in human keratinocytes [2], it is not sufficient to produce a fully transformed phenotype, suggesting that other genetic changes participate in the development of cervical cancer [3].

MicroRNAs (miRNAs) are small non-coding RNAs 21–25 nucleotides (nt) in length grouped in families with similar, as well as specific and unique, targets associated with cellular processes and molecular targets that remain largely undefined and experimentally untested [4,5]. Members of different miRNA families evolved to target a diverse set of transcripts via mechanisms including arm switching, seed shifts, insertions, and nucleotide editing of mature transcripts, generating different seed sequences and, hence, unique target specificities [6]. The members of different miRNA families present overlapping messenger RNA (mRNA) targets and tissue specificity [7]. Interestingly, miRNAs from a family do not express equally, neither do the 5p and 3p strands of a pre-miRNA [8,9,10]. Mature miRNA biogenesis begins with primary transcript (pri-miRNA) cleavage into an approximately 70-nt-long hairpin precursor miRNA (pre-miRNA) via the Drosha and DiGeorge Syndrome Critical Region Gene 8 (DGCR8) complex [11,12]. The pre-miRNA is exported to the cytoplasm by Exportin-5 [13,14], where a second processing step catalyzed by Dicer produces a miRNA duplex intermediate [15]. Single-strand mature miRNAs have the coding potential to hybridize with very different complementary mRNA targets.

MicroRNA strand maturation is a complex process consisting of two cuts, sequence thermodynamic instability, and protein interactions. Formation of 5p strand is carried out by the cleavage realized by microprocessor complex (Drosha and DGCR8) [16] accompanied by the interaction of p68, p72, Smads, p53, and ERα proteins. Other proteins directly interact with the structure and sequence of pri-miRNA and/or pre-miRNA, like hnRNP A1 and KSRP that bind to and increase pri-miRNA and/or pre-miRNA dicing contrary to the Lin 28 mechanism that inhibits the miRNA cleavage [17]. The 3p strand is formed by Dicer dicing and its activity is favored by Trans-Activation Responsive TAR RNA binding protein 2 (TRBP) and protein kinase, interferon-inducible double-stranded RNA-dependent activator (PACT) interaction. Protein post-translational modifications contribute to miRNA cropping. Phosphorylation of TRBP via Mitogen-Activated Protein Kinase (MAPK) increases Dicer enzymatic function, as well as hydroxylation by type I collagen prolyl-4-hydroxylase, which stabilizes Ago 2 [18]. MicroRNA strand selection is dependent on cell type, tissue, and stimulus [16]; therefore, sequence conservation and RNA-binding proteins evolved together to achieve a final fine-tuning of gene regulation. MicroRNA unbalanced expression conduces to oncogenic microRNA (oncomiR) or anti-oncomiR formation displaying tumor anti-apoptotic and proliferation induction or proliferation suppressive and pro-apoptotic functions, respectively [4]. The members of the miR-34 family are well recognized as anti-oncomiRs [19,20].

Several studies reported miR-34 family member dysregulation in various solid tumors including cervical cancer [21,22,23,24]. In high-risk HPV-infected cells, the interference of p53 and retinoblastoma protein (pRB) functions via the viral oncoproteins E6 and E7 disrupts the expression of hundreds of genes associated with cell-cycle regulation, DNA repair, and apoptosis. The overall effect of E6/E7 expression is reflected in genomic instability, unrestricted E2F bypass of the gap 1–synthesis (G1–S) checkpoint, and inhibition of p53-mediated functions including apoptosis [25,26,27,28,29]. The miR-34 family is composed of three evolutionarily conserved miRNAs: miR-34a, miR34b, and miR-34c expressing 5p strands (guide strand) and 3p strands or strand* (passenger strand). All three are described as direct targets of p53 and possess anti-proliferative and pro-apoptotic functions [10,30,31,32,33]. Specifically, miR-34a is the best characterized family member and is located within exon 2 of its primary transcript, while miR-34b miR-34c are produced from a different locus and positioned within intron 1 and exon 2 from the single primary transcript, respectively. Aside from the miRNAs themselves, the only other region of significant sequence conservation in miR-34 genes lies in their putative promoter regions harboring p53-binding sites [10]. Since high-risk HPV E6 oncoprotein destabilizes p53, a downregulation of the whole miR-34 family expression may be expected in HPV-positive cells. It was demonstrated that miR-34a is downregulated by E6 and the overexpression of miR-34a induces cell-cycle arrest and inhibits proliferation in cervical carcinoma cells [20].

Each member of miR-34 family has two identified mature miRNAs: miR-34a-5p, miR-34a-3p, miR-34b-5p, miR-34b-3p, miR-34c-5p, and miR-34c-3p [34]. The 5p strands of miR-34a, b, and c inhibit proliferation, anchorage-independent growth, migration, and invasion, and induce apoptosis in a variety of models; thus, it may be considered as a tumor suppressor [10,30,31,35]. The 5p strand of miR-34 members downregulates transcription factors (E2F3, MYCN), antiapoptotic genes B-cell lymphoma 2 (Bcl-2), oncogenic tyrosine kinase receptors (c-Met), AXL receptor tyrosine kinases (Axl), and cyclin dependent kinase 6 (CDK6) [35,36,37,38,39], among others (Appendix A). Although 5p members of the miR-34 family are well studied, and their function in cell processes such as proliferation, apoptosis, migration, and invasion is well accepted, the role of the 3p strand, as well as its targets, is largely unknown.

Here, we report that the overexpression of 5p and 3p strands of miR-34 family members inhibits proliferation, migration, and invasion in the HPV-positive tumor cell lines SiHa, CaLo, and C4.1 to different extents. Target prediction analysis in RNA22-HSA, TargetScan, and microRNA.org shows microtubule-associated protein 2 (MAP2), matrix metalloproteinase 9 (MMP9), and MMP2 as targets of miR-34 members. Tumor cells use matrix metalloproteinases 2 and 9 (MMP2 and 9) during invasion causing tumor dynamic changes [40]; additionally, MAP2 is a major regulator of microtubule dynamics in neuronal cells [41] and other cell types [42], and is associated with cell migration in epithelial cells [43]. Taking into consideration that these genes are directly or closely related in migration and invasion, we studied their expression under the overexpression of 5p and 3p strands of miR-34 family members. Interestingly, the transfection of an miR-34c-3p mimic led to the clear downregulation of MAP2 protein, as well as of MMP9 activity. Our findings suggest a possible antioncogenic regulating role of 5p and 3p strands, and provide evidence of the possible involvement of MAP2, MMP2, and MPP9 in cervical cancer regulated by miR-34a-5p, miR-34a-3p, miR-34c-5p, and miR-34c-3p.

## 2. Results

### 2.1. The miR-34 Family Members Inhibit Cervical Cancer Cell Proliferation

SiHa, CaLo, and C4.1 cells transfected with pre-miR-34a-5p, pre-miR-34a-3p, pre-miR-34b-5p, pre-miR-34b-3p, pre-miR-34c-5p, and pre-miR-34c-3p mimics resulted in a differential and significant decrease in proliferation compared with non-treated (NT), mock, and control (C)-treated cells measured by a reduction in crystal violet signal 72 h post-transfection (*p* < 0.05). The inhibition was considered specific to miR-34 members because controls did not show a significant reduction in proliferation (Figure 1A).

SiHa cell transfection with miR-34a-5p and miR-34a-3p recorded a cell proliferation inhibition of 38.4% and 33.8%, respectively, while miR-34b-5p showed 48.8% and miR-34b-3p showed 32.1% proliferation inhibition. Furthermore, miR-34c-5p and miR-34c-3p transfection showed 53.4% and 72.7% inhibition compared with controls as previously demonstrated [19]. The order of cell proliferation inhibition was as follows: miR-34c-3p, miR-34b-5p, miR-34c-5p, miR-34a-5p, miR-34a-3p, and miR-34b-3p (Figure 1A). CaLo transfected cells showed a similar effect with miR-34a-5p and miR-34b-5p, and miR-34c-5p and miR-34c-3p, while a lesser effect with miR-34b-5p and miR-34b-3p was recorded (Figure 1B). In C4.1 transfected cells, miR-34a-5p and miR-34b-5p achieved a more potent effect (71% and 65.5%, respectively), while the remaining miR-34 members showed ~53% cell proliferation inhibition (Figure 1C).

In SiHa cells, miR-34c-3p was the most potent, while, in CaLo cells, there was no significant difference between arms, and, in C4.1 cells, miR-34a-5p and miR-34b-5p had the greatest proliferation inhibition (Figure 1). Therefore miR-34 family members potentially regulate differential and specific targets to achieve cell proliferation inhibition.

### 2.2. The miR-34 Family Members Inhibit Migration and Invasion in SiHa Cells

Increased migration, metastasis, proliferation, and anchorage-independent growth, along with reduced senescence, angiogenesis, and inhibited apoptosis, are all cancer hallmarks [42]. As mentioned above, SiHa cells presented the most potent proliferation inhibition effect with miR-34 family members; therefore, the effect on migration and invasion by miR-34 family members in SiHa cells was analyzed. Transfection of the pre-miR-34 family member mimics on SiHa cells inhibited migration and invasion relative to NT, mock, and C-treated cells (Figure 2A,B).

The migration assay showed that all the members of the miR-34 family inhibit cell motility. Noteworthy, the extent of inhibition was different with all strands in the following order: miR-34c-3p, miR-34a-3p, miR-34a-5p, miR-34b-5p, miR-34b-3p, and miR-34c-5p (Figure 2A). Overall, miR-34a-3p induced a lesser effect in invasion (Figure 2B), contrary to the migration assay (Figure 2A). Interestingly, miR-34a-5p, miR-34b-3p, and miR-34c-5p induced a similar effect in migration and invasion (Figure 2A,B), while miR-34a-5p had the most potent inhibitory effect on invasion (Figure 2B). Furthermore, despite miR-34a-3p, miR-34b-5p, and miR-34c-3p inhibiting invasion, the comparison between migration and invasion showed a different pattern (Figure 2A,B).

MMP2 and MMP9 are well-recognized markers of invasion [44,45]; therefore, we analyzed the activity of the two metalloproteases. We searched for binding sites of miR-34 family members in MMP2 and/or MMP9 mRNAs, and we found that miR-34a-5p and miR-34c-5p had binding sites for these genes (Figure 2C). However, miR-34a-5p and miR-34c-5p that have 3’ untranslated region (UTR) binding sites did not show any effects, contrary to the miR-34a-3p mimic that showed a reduction in MMP2 activity despite 3’ UTR binding sites being absent (Figure 2D,E). Moreover, miR-34a-5p, miR-34b-5p, miR-34b-3p, miR-34c-5p, and miR-34c-3p overexpression did not present a statistically differential effect in MMP2 activity (Figure 2E). On the other hand, MMP9 activity was diminished by miR-34a-5p, miR-34b-3p, miR-34c-5p, and miR-34c-3p (Figure 2G,H). In accordance with database RNA22-HSA MMP9 is targeted by miR-34a-5p and miR-34c-5p (Figure 2C); interestingly, miR-34b-3p and miR-34c-3p reduced its activity. It is not clear if the 3p strands of miR-34b and c target MMP9 directly or indirectly.

As MAP2 is associated with migration in epithelial cells [43], we analyzed the effect of miR-34 family members on its expression. MAP2 has binding sites for all miR-34 family members (Figure 2F), although only miR-34a-5p, miR-34c-5p, and miR-34c-3p downregulated MAP2 in a statistically significant manner (Figure 2G,H). The role of MAP2 in cell migration in cervical cancer is not well understood; however, it could be one of the genes that miR-34 family members regulate to achieve their function in the cells. The down- and/or upregulation of MAP2 observed in this work needs further studies. Migration and invasion are not totally explained by MMP9, MMP2, and MAP2 protein expression inhibition; however, it must be taken into consideration that every strand has several mRNAs targets that could be involved in cellular processes.

### 2.3. In Silico Analysis of Targets Regulated by miR-34 Family Members

The effect of the miR-34 family could not be well explained with the evaluation of three genes; therefore, we explored three databases to see if at least the members of miR-34 family have potentially unique genes that could elucidate the differential effects observed. RNA22-HSA, DIANA-microT, and miRDB showed a different quantity of genes (Table 1), while the 5p arms presented more targets than the 3p arms (Figure 3A,B); therefore, it should be expected that the 5p and/or 3p arms of miR-34a, b, or c present a similar effect. In CaLo cells, the percentage of potential mRNA-regulated targets was similar (Figure 3A,B) to the proliferation effect of miR-34b-5p (41%) and miR-34b-3p (34.7%) (Figure 1B). As it can be seen in SiHa cells, miR-34c-3p presented the most potent effect (Figure 1A), while, in C4.1, miR-34a-5p and miR-34b-5p were the strongest inhibitors (Figure 1C). Interestingly, the 5p arms of miR-34 family members were found to potentially regulate more targets than 3p arms. However, in SiHa cells, miR-34c-3p was the most potent, while, in CaLo cells, there was no significant difference between arms, and, in C4.1 cells, miR-34a-5p and miR-34b-5p had the greatest proliferation inhibition (Figure 1) that could be explained by the 5p and 3p strands individually regulated targets.

For example, miRDB miR-34a-5p potentially regulates 12 genes individually and miR-34b-5p regulates 316 genes, while miR-34c-5p regulates three individual genes (Table 1). The 5p arms of miR-34a, b, and c regulate 76 genes in common (Figure 3A). Individually, miR-34a-3p potentially regulates 217 genes, miR-34b-3p regulates 250 genes, and miR-34c-3p regulates 128 genes; it could be noted the 3p arms regulate more genes individually (Figure 3B), despite the 5p arms potentially regulating more targets (Table 1). The same pattern was shown in the RNA22-HSA and miRDB databases (Figure 3A,B; Table 1).

The genes were ranked in the following cell and molecular processes: nuclear factor (NF) κB signaling pathway, galactose metabolism, chemical carcinogenesis, Gonadotropin-Releasing Hormone Receptor (GnRH) signaling pathway, fatty-acid degradation, and gamma-aminobutyric acid (GABA)ergic synapse, among others (Table 2). The cell-signaling pathways were selected based on the exclusivity of each miR-34 family member strand. We grouped cell-signaling pathways that are potentially regulated by 5p and 3p strands of miR-34 family members into six pathways. We found that miR-34a-5p directly induces apoptosis via Blc-2 inhibition and indirectly via downregulation of PRKCB/IkBα (Figure 3C). Bcl-2 and PRKCB appear as validated targets for miR-34a-5p in DIANA, miRanda, miRbridge, Pictar, RNA22HSA, and TargetScan databases, while VGEF was not validated. Furthermore, miR-34b-5p presents a dual effect on growth suppression; on one hand, it increases suppression via MYC and, on the other hand, it decreases suppression via extracellular signal-regulated kinase 2 (ERK2) and Ras-related Protein R-Ras2 (RRAS2). MYC and ERK2 were validated as targets for miR-34b-5p in Pictar and RNA22-HAS, while RRAS2 was not. We also found that miR-34c-5p diminishes cell proliferation and invasion via Wnt/β-cateninTCF/LEF. In databases DIANA, miRanda, miRbridge, Pictar, RNA22HSA, and TargetScan, Wnt does not appear as a valid target for miR-34c-5p. Moreover, miR-34a-3p inhibits cell-cycle progression and proliferation via growth factors such as epidermal growth factor, platelet-derived growth factor subunit A (PDGFA), platelet-derived growth factor subunit B (PDGFB), platelet-derived growth factor subunit C (PDGFC), and platelet-derived growth factor subunit D (PDGFD), as well as Mitogen-Activated Protein Kinase 8 (MAPK8), PI3K, AP-1, and phospholipase D1 (PC-PLD-1). In databases DIANA, miRanda, miRbridge, Pictar, RNA22, and TargetScan, PI3K, FOS, MAP8, and PC-PLD1 do not appear as a validated targets for miR-34a-3p. We also found that miR-34b-3p inhibits cell growth differentiation via MET, and TAK1 modulating Wnt/β-catenin. MET appears as a validated target for miR-34b-3p in the DIANA, miRanda, and PITA databases, while TAK1 was not validated as a target for miR-34b-3p in the databases consulted. Finally, miR-34c-3p inhibits cell proliferation and differentiation via PDGFB and Ras guanyl-releasing protein 1 (RASGRP) controlling MAPK signaling (Figure 3D). RASGRP appears as a miR-34c-3p validated target in the DIANA database, while PDGFB was not validated in the databases consulted. The genes participating in the six cell-signaling pathways potentially regulated by 5p and 3p strands of miR-34 family members are overexpressed in cervical cancer [46,47,48,49,50,51,52,53]. The genes encoding VGEF, RRAS2, PRKCB, RASGRP, and PDGFB involved in the six cell-signaling pathways are not currently reported in cervical cancer; however, they were reported as overexpressed in other carcinomas [54,55,56,57,58], suggesting their participation in carcinogenesis.

The 5p strand presents more targets in databases RNA22-HSA, miRDB, and DIANA-microT and, therefore, more potential regulated targets in common, while the 3p arms have fewer targets, as well as fewer potentially regulated targets (Figure 3A,B). In contrast, the 3p arms potentially regulate more individual targets than the 5p arms (Table 2).

## 3. Discussion

It is well documented that most human cancers show a global decrease in mature miRNA expression, resulting in cellular transformation and tumorigenesis [59,60]. Deficiencies in p53 result in the downregulation of miR-34 family expression, which is, in turn, associated with an increase in proliferation and colony formation, cell-cycle progression, and senescence, as well as apoptosis inhibition [35,61]. Although it was shown that miR-34a, miR-34b, and miR-34c have distinct or overlapping targets, it is only beginning to be established which of these mRNA targets harbors their anti-proliferative effects. Previous work on miR-34a, b, and c 5p strand overexpression showed cell proliferation reduction, senescence, apoptosis induction, and gap 1 (G1) phase arrest in several cell lines through the inhibition of E2F3, Bcl-2, and c-Myc [33,38,62]. In this work, we analyzed the function of 5p and 3p strands of miR-34 family members function in cervical cancer cell lines. In the present study, ectopic expression of 5p or 3p strands inhibited SiHa, C4.1, and CaLo cell proliferation to different extents, as well as migration and invasion in SiHa Cells. MMP2 and MMP9 activity, and the protein expression of MAP2 were differentially regulated by 5p and 3p strands of miR-34 family members.

The differential activity of the two possible mature miRNAs for a single pre-miRNA hairpin was reported for other miRNAs such as miR199a [63], miR-17 [64], and miR-125a [65], among others. Nevertheless, the activities of 5p arms in SiHa, CaLo, and C4.1 cells appear redundant rather than complementary or antagonistic to the 3p passenger arm. It was found that miR-34a-5p, miR-34c-5p. and miR-34c-3p diminish MAP2 protein expression; however, its different regulation by miR-34 members does not explain the differences between cellular processes. MAP2 is a protein associated with microtubule stabilization and an important marker for neuronal differentiation. In addition, the participation of MAP2 in binding to filamentous actin, recruitment of signal proteins, and regulation of microtubule-mediated transport was described [66]. MAP2 was shown to be overexpressed in cutaneous melanoma [67] and is considered a specific marker for pulmonary carcinoid tumor and small-cell carcinoma [68]; however, its biological function in cervical cancer tumorigenesis is poorly understood. MAP2 expression is correlated with cell motility and invasion in epithelial and oral cancer cells [43]. Accordingly, we report here the endogenous MAP2 expression in cervical carcinoma cells, the downregulation of which was produced by the ectopic expression of 5p and 3p strands of miR34a and c in SiHa cells, respectively. Although MAP2 involvement in cell motility remains unclear, it can serve as an architectural element by establishing specific morphological features and specific arrangements of the microtubules [69]. Essential roles of microtubules include segregating chromosomes and positioning the nucleus during mitosis. Defects in these functions can lead to proliferation inhibition and cell death in tumor cells [70]. Microtubules are important factors involved in chromosome aneuploidy, which account for cell mutation that can lead to or enhance metastasis. Therefore, the observed proliferation inhibition, migration, and invasion may all be a consequence of microtubule dysfunction produced by the miR-34 family. Nevertheless, the microtubule organization of miR34 family-expressing cervical carcinoma cells remains to be tested. A regulatory interaction between MAP2 and invasion in cervical carcinoma cells is yet to be demonstrated, although the functions of both in cell motility/invasion may suggest it. It was demonstrated that MMP2 and MMP9 participate in cell invasion; miR-34 family members inhibit MMP2 and MMP9 activity differentially given the high complexity and specificity of this family. Remarkably, miR-34a-3p uniquely inhibits MMP2 activity in contrast with its null effect on invasion. It could be possible that miR-34a-3p directly or indirectly downregulates MMP2 activity. It must be observed that MMP2 does not have binding sites for the 3p strand of miR-34a-3p; nevertheless, several miRNAs regulate mRNAs independently of the seed sequence [71].

MMP9 activity is reduced by miR-34a-5p, miR-34b-3p, miR-34c-5p, and miR-34c-3p. In SiHa and HeLa cells, MMP2 and MMP9 activity reduction by miR-183, miR-125a, miR-491-5p, and miR-454-3p was reported [72,73,74,75]; nevertheless, for miR-34 family members in cervical cell lines, there are no data. However, MMP2 and MMP9 are inhibited by miR-34a-5p in esophageal squamous cell carcinoma [48]. The differential effect between cervical cancer cell lines and esophageal squamous cell lines could be a reflection of the cellular context of the gene environment, highlighting that MMP2 and MMP9 are downregulated by several miRNAs, independent of UTR binding sites [73,74,75]. Additionally, MMP2 and MMP9 could be regulated indirectly via their activators (MT1-MMP and heregulin-B1, plasmin, U-Plasminogen Activator (uPA), MMP-3, MMP-2, and MMP-13) [76,77,78,79] and/or inhibitors (Tissue Inhibitor of Metalloproteinase (TIM-2 and TIM-1)) [79,80].

Messenger RNA targets for 5p and 3p arms could explain the difference in proliferation, migration, and invasion. The 5p strand has more targets than the 3p strand, probably as a result of sequence conservation. The 3p strand presents more individually regulated targets, probably giving specificity to gene regulation based on sequence conservation [81]. It should be noted that the 5p strand is present in the cell three times more often than the 3p strand [82]. It could be possible that thousands of genes regulated by the 5p strand define the fate of cellular processes and that, upon 3p strand expression, the cellular fate changes dramatically, as observed with miR-34c-3p in proliferation, migration, and invasion assays. Additionally, the diverse mechanisms regulating miRNA-mediated gene silencing may account for the difference in the effect of miR-34 family members. Considering the influence of the cell background in the function of a particular miRNA [83], nucleotide changes within the target seed sequence, the interaction with 3’ UTR binding proteins, and/or the abnormal activation of the miRNA regulatory networks could all reflect an escape from normal miRNA regulation in tumor cells. The targets achieved by miRNAs could influence their instability, affecting and/or influencing the level of cell process inhibition [84,85]. Cell-signaling pathways PRKC/IkBα/Bcl-2, Wnt/β-catenin/TCF/LEF, VGEF/TGFβ, MYC/p21/Waf-1, CD44/ERα/CCND1, and Grb2/ERK are regulated by 5p strands of miR-34 family members, while cell-signaling pathways GF/ERK1, GF/RTK/SRF/c-Fos, MET/TCF/LEF, TAK1/NLK, GF/RTK/PI3K/PLD1, and AP1/CHK are regulated by 3p strands of miR-34 family members. Based on 10 pathways exclusive to each strand, we selected genes based on their repetition in more than three pathways. Interestingly, TCF/LEF and ERK are shared between miR-34 family member strands. This prediction needs further validation pointing out the complexity of cell regulation by 5p and 3p strands. The targets individually regulated by each strand could explain the differences between proliferation, migration, and invasion observed in this work; however, the specific mRNA targets regulated by 5p and 3p arms of miR-34 family members are needed to elucidate the different effects on cell processes. Nevertheless, the powerful growth inhibition function of miR-34a-5p, miR-34c-5p, and miR-34c-3p in SiHa cells represents a first step toward the identification of an important target with potential application in therapy and prognosis of cervical cancer.

## 4. Materials and Methods

### 4.1. Cell Culture

The cervical tumor lines SiHa, CaLo, and C4.1 were kindly donated by Dr. Patricio Gariglio from CINVESTAV-IPN, Mexico. Cells were grown in Dulbecco’s modified Eagle’s medium (DMEM) (Invitrogen Corporation, Carlsbad CA), enriched with 5% fetal bovine serum (FBS) (Invitrogen), and cultivated at 37 °C and 5% CO_2_ in a water-saturated atmosphere.

### 4.2. Transient miRNA Transfection

Pre-miRNA mimics (AM1700-IDPM11030, AM1700-IDPM13089, AM1700-IDPM10743, AM1700-IDPM12727, AM1700-IDPM11039, and AM1700-IDPM12342) and a scrambled pre-miRNA negative control (AM17121) were purchased from Applied Biosystems Ambion Inc., Austin TX. For transient transfection, pre-miRNAs and 1.5 µL of Lipofectamine™ 2000 reagent (Invitrogen) were diluted with 100 µL of basal DMEM, and mixed to reach 211.5 µL of pre-miRNA/Lipofectamine 2000 complexes. Cells were trypsinized, counted, and resuspended in 500 μL of 1× phosphate-buffered saline (PBS). The pre-miRNA/Lipofectamine mixtures were added directly to 1 × 10^5^ suspended cells, and were seeded in six-well dishes before being adjusting to 600 µL with basal DMEM. The cells with pre-miRNA/Lipofectamine mixtures were incubated and shacked for six hours, and 400 µL of 5% DMEM was added to adjust the final volume to 1 mL. After overnight culture, the transfection mixture was removed, and 3 mL of DMEM supplemented with 15% FBS was added. Mock-transfected cells received identical treatment except for the addition of pre-miRNAs. Data were subjected to a two-tailed Student’s *t*-test. A *p*-value <0.05 was considered as statistically significant. Graphs were plotted using the means and standard errors from three independent experiments.

### 4.3. Cell Proliferation Analysis

Cell proliferation was quantified using violet crystal dye in 1× phosphate-buffered saline (PBS) (2.7 mM KCl, 1.8 mM KH_2_PO_4_, 136 mM NaCl, 10 mM Na_2_HPO_4_; pH 7.4). The transfected cells were incubated in methanol for 15 minutes and washed two times with water. Cells were dyed with 0.1% crystal violet and washed three times with water. Finally, crystal violet was recovered with 10% acetic acid to be analyzed at 600 nm optical density in a microplate reader (Multiskan GO Spectrophotometer, Thermo Scientific™).

### 4.4. Cell Migration and Invasion Measured by Transwell Assays

Cells were transfected with pre-miRNA mimics (5 nM) and treated with mitomycin C (1.2 µg/mL) for 1.5 h. They were subsequently trypsinized, counted (8 × 10^4^ cells/transwell), and seeded into a transwell chamber (upper compartment), before adjusting the final volume to 200 µL of DMEM supplemented with 2% FBS. As a chemoattractant, 600 µL of DMEM supplemented with 10% FBS was used in the lower compartment, which was incubated for 72 h; the cells were washed with PBS, and fixed in 100% methanol. The non-invading cells were removed from the upper surface of the membrane by scrubbing, followed by staining with 0.1% crystal violet, and they were photographed with an AE 2000 inverted microscope (Motic^®^). Afterward, the membranes were incubated with 10% acetic acid and measured in a spectrophotomer at 600 nm optical density. Invasion assays were analyzed similarly to the migration assay, except that the insert was covered with matrigel at 0.1% before 8 × 10^4^ cells/transwell were seeded.

### 4.5. Zymography

For the analysis of the activities of MMP2 and MMP9 in the supernatant of SiHa cells, 5 µL of supernatant was placed and mixed with 2.5 mL of run buffer (2.5% SDS, 1% sucrose) in a 7.5% SDS polyacrylamide gel containing 1 mg/mL gelatin (Sigma-Aldrich) without heating and addition of β-mercaptoethanol. Electrophoresis was run at a constant 90 V until the dye reached the bottom of the gel. The gels were washed twice in 2.5% Triton-X 100 for 20 minutes and were incubated for 48 hours at 37 °C in an assay buffer consisting of 50 mM Tris-HCl (pH 8.0) and 2 mM CaCl, before being stained with 0.025% (*w*/*v*) Coomassie Blue R250 for 1 h with gentle shaking. The gels were destained for one hour in 8% acetic acid and 4% methanol. The gelatinolytic activity of the MMP bands was quantified by densitometric analysis (Quantity One program, Gel Doc 2000 system, Bio-Rad, USA).

### 4.6. Immunoblotting

Cells were extensively washed with 1× PBS and lysed in radioimmunoprecipitation assay (RIPA) buffer (50 mM Tris-HCl pH 8.0, 150 mM NaCl, 10 mM sodium deoxycholate, 1% (*v*/*v*) NP-40, 0.1% (*w*/*v*) SDS, 0.5 mM 4-(2-aminoethyl) benzenesulfonyl fluoride hydrochloride (AEBSF) and 4-(2-aminoethyl) benzenesulfonyl fluoride hydrochloride, Pepstatin A, Bestatin hydrochloride, E-64 protease inhibitor, and Leupeptin (Sigma Aldrich^®^). Samples were quantified using the Bradford method [35]. Firstly, 50 μg of total soluble protein was denatured in Laemmli loading buffer (62.5 mM Tris-HCl pH 6.8, 25% glycerol, 2% SDS, 2% (*v*/*v*) β-mercaptoethanol, and 0.01% (*w*/*v*) bromophenol blue) and boiled for 5 min before gel electrophoresis using 10% polyacrylamide/SDS. Gels were electro-transferred to a HYBOND™-C extra supported nitrocellulose membrane (GE Healthcare Life Sciences) for 2 hours at 300 mA. A nitrocellulose membrane was blocked in 5% nonfat milk powder in 1% PBS with Tween20 at room temperature for 2 h and incubated with primary antibodies for MAP2 (sc-20172), and β-actin (sc-1616) (Santa Cruz Biotechnologies, Santa Cruz CA) overnight. Primary and secondary antibodies were used at dilutions of 1:1000 and 1:10000, respectively. Secondary antibodies were incubated one hour at room temperature. Detection was carried out using Western Blotting Luminol Reagent sc-2048 (Santa Cruz Biotechnology, Inc.)

### 4.7. Messenger RNA Target Prediction Analysis in RNA22-HAS, DIANA-microT, and miRDB Databases

Target genes for miR-34 family members were obtained from miRDB (http://mirdb.org), RNA22-HSA (https://cm.jefferson.edu/rna22/Precomputed/), and DIANA microT-CDS (http://www.microrna.gr/microT-CDS). The data were saved in .xlsx and .txt format in Excel and were imported to Wolfram mathematics and Venn. The genes that were obtained from this analysis were introduced to Enrichr [46] to determine pathways regulated by each strand of miR-34 family members.

## 5. Conclusions

The present data provide evidence that the 3p strand of miR-34 family members has a function in cell proliferation, cell migration, and invasion in SiHa, CaLo, and C4.1 cells in a different manner to the 5p strand. We found that miR-34a-5p has a remarkable effect in cell migration and invasion that, in part, could be explained by MMP9 activity inhibition and downregulation of MAP2 in SiHa cells; however, MMP2 was not affected by this miRNA. Furthermore, miR-34a-3p, miR-34b-5p, miR-34b-3p, and miR-34c-5p inhibit proliferation and migration, but not invasion; this could, in part, be attributable to MMP2 and MMP9 activity inhibition and MAP2 protein downregulation. Moreover, miR-34a-3p and miR-34c-5p downregulate MMP2 activity and MAP2 protein, respectively, while miR-34b-3p inhibits MMP9 activity, highlighting the participation of other genes involved in invasion in addition to MMP2 and MMP9. Finally, miR-34c-3p inhibits proliferation, migration, and invasion, which could be attributable to MMP9 activity inhibition and MAP2 protein expression reduction. Nevertheless, given the difference in cellular processes (MMP2 and MMP9 activity, and MAP2 protein expression inhibition), the effect of the miR-34 family members is a reflection of other regulated genes. Further experiments are needed to elucidate the mechanisms underlying the differential effect in cervical carcinoma cell lines.

## Figures and Tables

**Figure 1 ijms-20-00545-f001:**
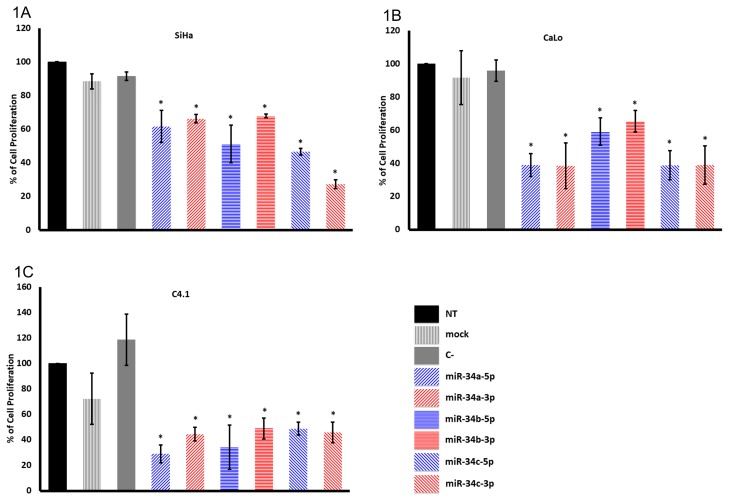
Ectopic expression of microRNA 34 (miR-34) family members inhibits proliferation in SiHa, CaLo, and C4.1 cells. (**A**) The human papillomavirus (HPV)-16-positive tumor cell line SiHa; (**B**) the HPV-18-positive tumor cell line CaLo; (**C**) the HPV-18-positive tumor cell line C4.1. The cell lines were transfected with 10 nM pre-miR-34a-5p, pre-miR-34a-3p, pre-miR-34b-5p, pre-miR-34b3p, pre-miR-34c-5p, and pre-miR-34c-3p mimics, or scrambled pre-miRNA control (C-) to evaluate cell proliferation with crystal violet 72 h post-transfection. Non-treated (NT) and mock-transfected (mock) cells were used as positive proliferation controls. The bars represent means and standard deviations of three independent experiments in triplicate (*p* < 0.05).

**Figure 2 ijms-20-00545-f002:**
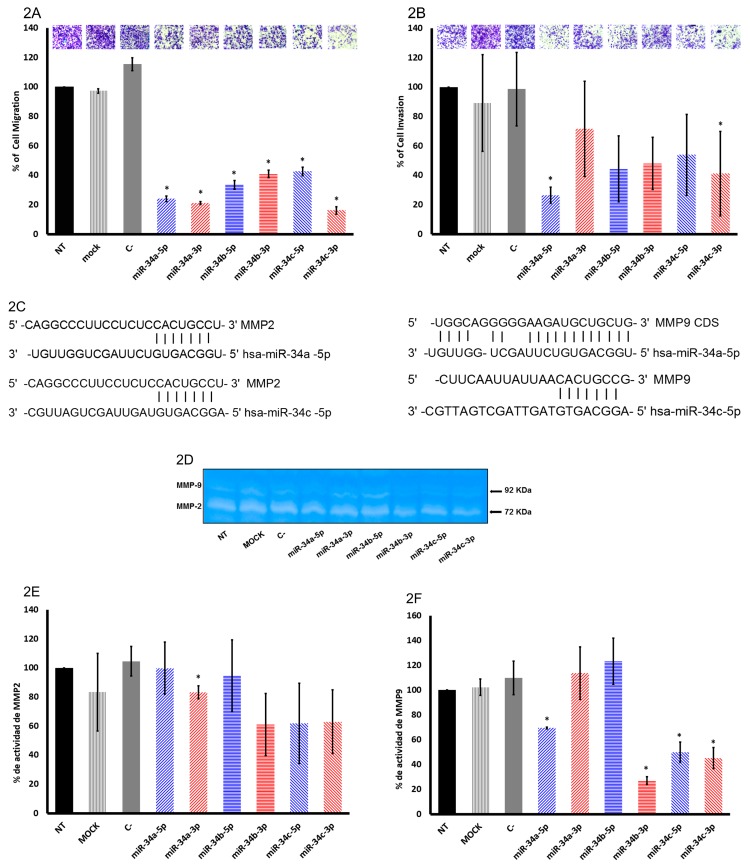
Ectopic expression of miR-34 family members affects cell migration and invasion in SiHa cells. (**A**) SiHa cells were transfected with 5 nM pre-miR-34a-5p, pre-miR-34a-3p, pre-miR-34b-5p, pre-miR-34b3p, pre-miR-34c-5p, and pre-miR-34c-3p mimics, or scrambled pre-miRNA control (C-) mimic. The cells were treated with mitomycin C (1.2 µg/ml) and 8 × 10^4^ cells were seeded in transwell inserts to analyze migration 72 hours post-transfection. (**B**) The inserts were recovered with matrigel and 8 × 10^4^ cells were seeded, previously transfected with 5 nM pre-miR-34a-5p, miR-34a-3p, miR-34b-5p, miR-34b3p, miR-34c-5p, and miR-34c-3p mimics, or scrambled pre-miRNA control (C-) mimic and treated with mitomycin C (1.2 µg/ml) to analyze invasion 72 hours post-transfection. The photograph on the top corresponds to one representative experiment of the assays. (**C**) Schematic representation of miR-34 family members binding sites in the 3’ untranslated region (UTR) of matrix metalloproteinases 2 and 9 (MMP2 and MMP9) with databases miRWalk, Target Scan, and RNA22-HSA. (**D**) Supernatant was collected from cells transfected with 10 nM pre-miR-34a-5p, miR-34a-3p, miR-34b-5p, miR-34b3p, miR-34c-5p, and miR-34c-3p mimic, or scrambled pre-miRNA control (C-) mimic, and MMP2 and 9 activity was analyzed 72 hours post-transfection. (**E**) Graphs showing the quantification of the effect of 5p and 3p strands of miR-34 family members on MMP2 activity from three independent experiments (*p* < 0.05). (**F**) Graphs showing the quantification of the effect of 5p and 3p strands of miR-34 family members on MMP9 activity from three independent experiments (*p* < 0.05). (**G**) Schematic representation of miR-34 family members binding sites in the 3’ UTR of microtubule-associated protein 2 (MAP2) with databases miRWalk, Target Scan, and microRNA.org. (**H**) Protein extraction was obtained from cells transfected with 10 nM pre-miR-34a-5p, miR-34a-3p, miR-34b-5p, miR-34b3p, miR-34c-5p, and miR-34c-3p mimics, or scrambled pre-miRNA control (C-) mimic, and MMP2 activity was analyzed using Western blot. (**I**) Graph showing the quantification of the effect of 5p and 3p strands of miR-34 family members on MAP2 protein from three independent experiments in triplicate (*p* < 0.05). The bars represent the means and standard deviations of three independent experiments (*p* < 0.05).

**Figure 3 ijms-20-00545-f003:**
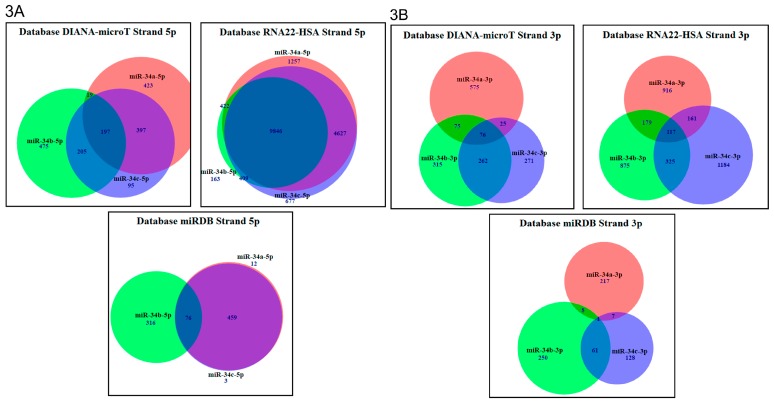
Potential targets of miR-34 family members searched on miRDB, DIANA-microT, and RNA22-HSA. (**A**) Messenger RNA (mRNA) targets of 5p strand of miR-34 family members were analyzed from data obtained in miRDB, DIANA-microT, and RNA22-HSA, and genes were grouped as common or individually regulated. (**B**) Targets of the 3p strand of miR-34 members were analyzed in miRDB, DIANA-microT, and RNA22-HSA, and genes were grouped as common or individually regulated. (**C**) Cell-signaling pathways Protein Kinase C beta (PRKC)/ NF-κB inhibitor α (IkBα)/Bcl-2 inhibiting apoptosis, Wingless-type MMTV integration site family (Wnt)/ catenin (cadherin-associated protein), and beta 1 (β-catenin)/ T-cell factor (TCF)/ Lymphoid Enhancer Binding Factor (LEF) inducing proliferation and invasion, Vascular Endothelial Growth Factor (VGEF)/ Transforming Growth Factor Beta 1 (TGFβ) inducing angiogenesis, MYC/p21/Waf-1 inducing growth suppression, and Cluster Differentiation 44 (CD44)/ Estrogen Receptor Alpha (ERα)/Cyclin D1 (CCND1) and Growth Factor Receptor Bound Protein 2 (Grb2)/ Extracellular Signal-Regulated Kinase (ERK) stimulating cell growth and survival are regulated by 5p strands of miR-34 family members. (**D**) Cell-signaling pathways Growth Factor (GF)/ Extracellular Signal-Regulated Kinase 1 (ERK1) stimulating cell growth and survival, GF/Receptor Tyrosine Kinase (RTK)/Serum Response Factor (SRF)/c-Fos inducing proliferation and differentiation, MET/TCF/LEF inducing cell growth and differentiation, Transforming Growth Factor-Beta-Activated Kinase 1 (TAK1)/ Nemo Like Kinase (NLK) inhibiting cell growth and differentiation, GF/RTK/ Phosphatidylinositol-4,5-Bisphosphate 3-Kinase (PI3K)/ Phospholipase D1 (PLD1) inducing proliferation, cell-cycle progression, and survival, and Activator Protein 1 (AP1)/Checkpoint Kinase (CHK) stimulating proliferation, cell-cycle progression, and survival are regulated by 3p strands of miR-34 family members.

**Table 1 ijms-20-00545-t001:** Genes regulated by 5p and 3p strands of microRNA 34 (miR-34) family members.

Data Base	Total Genes Regulated by 5p Strands	Total Genes Regulated by 3p Strands	Genes Regulated by 5p Strands	Genes Regulated by 3p Strands	% of Genes Regulated by 5p Strands	% of Genes Regulated by 3p Strands
miRDB	866	672	miR-34a (12) miR-34b (316) miR-34c (3)	miR-34a (217) miR-34b (250) miR-34c (128)	miR-34a (1.38%) miR-34b (36.48%) miR-34c (0.34%)	miR-34a (32.29%) miR-34b (37.2%) miR-34c (19.04%)
RNA22-HSA	17,401	3752	miR-34a (1257) miR-34b (163) miR-34c (677)	miR-34a (916) miR-34b (875) miR-34c (1184)	miR-34a (7.22%) miR-34b (0.93%) miR-34c (3.89%)	miR-34a (24.41%) miR-34b (23.32%) miR-34c (31.55%)
DIANA-microT	1811	1599	miR-34a (423) miR-34b (475) miR-34c (95)	miR-34a (575) miR-34b (315) miR-34c (271)	miR-34a (23.35%) miR-34b (26.22%) miR-34c (5.24%)	miR-34a (35.95%) miR-34b (19.69%) miR-34c (16.94%)

**Table 2 ijms-20-00545-t002:** Cell pathways regulated exclusively by 5 and 3p strands of miR-34 family members.

miRNA	Single Pathways by Strand
hsa-miR-34a-5p	DIANA microT
	-NF-κB signaling pathway -Longevity regulating pathway (multiple species) -Focal adhesion -Amphetamine addiction -Rap1 signaling pathway -Adrenergic signaling in cardiomyocytes -Pathways in cancer
	miRDB
	-Galactose metabolism -One-carbon pool by folate -Glycosaminoglycan biosynthesis-keratan sulfate -Butirosin and neomycin biosynthesis -Type II diabetes mellitus -Carbohydrate digestion and absorption -Glycosphingolipid biosynthesis, lacto and neolacto series -Amino sugar and nucleotide sugar metabolism -Other types of *O*-glycan biosynthesis
	RNA22-HSA
	-Chemical carcinogenesis -Steroid hormone biosynthesis -Metabolism of xenobiotics by cytochrome P450 -Parkinson’s disease
hsa-miR-34a-3p	DIANA microT
	-GnRH signaling pathway -Choline metabolism in cancer -Tryptophan metabolism -Dopaminergic synapse -Gastric acid secretion -Salmonella infection -Histidine metabolism -Chagas disease (American trypanosomiasis)
	miRDB
	-Pathways in cancer -Fatty-acid degradation -Histidine metabolism -Adherens junction -Tryptophan metabolism -Axon guidance -Tuberculosis -Apoptosis
	RNA22-HSA
	Pathways in cancer -GABAergic synapse -Circadian entrainment -Retrograde endocannabinoid signaling -PI3K/Akt signaling pathway -Morphine addiction -Dopaminergic synapse -Glutamatergic synapse -ErbB signaling pathway -Chemokine signaling pathway
hsa-miR-34b-5p	DIANA microT
	-Shigellosis -Aldosterone-regulated sodium reabsorption -Regulation of actin cytoskeleton -Proteoglycans in cancer -Pathogenic *Escherichia coli* infection -ErbB signaling pathway -Hepatitis B -Thyroid cancer
	miRDB
	-Ubiquitin-mediated proteolysis -Thyroid hormone signaling pathway -ErbB signaling pathway -Circadian rhythm -MicroRNAs in cancer -T-cell receptor signaling pathway -Prolactin signaling pathway -Insulin signaling pathway
	RNA22-HSA
	-Renin/angiotensin system -Lysosome -Nicotinate and nicotinamide metabolism -mRNA surveillance pathway -Influenza A -RNA transport -Fat digestion and absorption -Pyrimidine metabolism -Purine metabolism
hsa-miR-34b-3p	DIANA microT
	-Glycosaminoglycan degradation -mRNA surveillance pathway -Adherens junction -Glutathione metabolism -AMPK signaling pathway -SNARE interactions in vesicular transport -Transcriptional misregulation in cancer -Cysteine and methionine metabolism
	miRDB
	-Cysteine and methionine metabolism -Sphingolipid metabolism -Phospholipase D signaling pathway -Choline metabolism in cancer -Renin secretion -Synaptic vesicle cycle -Oocyte meiosis -Dorso–ventral axis formation
	RNA22-HSA
	-Type II diabetes mellitus -Phosphatidylinositol signaling system -Oxytocin signaling pathway -Protein digestion and absorption -AMPK signaling pathway -Butanoate metabolism -cAMP signaling pathway -Adrenergic signaling in cardiomyocytes -cGMP/PKG signaling pathway -Gastric acid secretion
hsa-miR-34c-5p	DIANA microT
	-Synaptic vesicle cycle -Vitamin B6 metabolism -Lysosome -Oxidative phosphorylation -Natural killer cell-mediated cytotoxicity -Glycosylphosphatidylinositol (GPI)-anchor biosynthesis -Circadian rhythm -Cocaine addiction
	miRDB
	-Chemokine signaling pathway -Cytokine–cytokine receptor interaction
	RNA22-HSA
	-Propanoate metabolism -Terpenoid backbone biosynthesis -Complement and coagulation cascades -Taste transduction -PPAR signaling Pentose phosphate pathway -Basal cell carcinoma -Asthma -Staphylococcus aureus infection
hsa-miR-34c-3p	DIANA microT
	-Biosynthesis of unsaturated fatty acids -Measles -Glutamatergic synapse -MAPK signaling pathway -T-cell receptor signaling pathway -Ubiquitin-mediated proteolysis -Renin secretion -Renal cell carcinoma -Central carbon metabolism in cancer -Inositol phosphate metabolism
	miRDB
	-Biosynthesis of unsaturated fatty acids -Maturity onset diabetes of the young -Pentose and glucuronate interconversions -Fatty-acid elongation -Endocrine and other factor-regulated calcium reabsorption -Fatty-acid metabolism -Regulation of actin cytoskeleton -Leukocyte transendothelial migration
	RNA22-HSA
	-Thyroid hormone signaling pathway -Retinol metabolism -Drug metabolism, cytochrome P450 -Signaling pathways regulating pluripotency of stem cells -Aldosterone-regulated sodium reabsorption

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
