# Peer review of "5p and 3p Strands of miR-34 Family Members Have Differential Effects in Cell Proliferation, Migration, and Invasion in Cervical Cancer Cells"

_ijms, 2019, doi:10.3390/ijms20030545_

Reviewer 1 Report

Córdova-Rivas et al. investigate the biological function of the miR-34a, b and c miRNAs 3p strands (the passenger strand) in cervical cancer cell lines. While the targets of the 5p strands are well characterized the 3p strand function and regulation remain to be elucidated. Using transient transfection, they study Cell proliferation was evaluated by crystal violet, Migration and invasion were tested by 36 transwell-assays, Western-blot and zymography. Predicted targets and cell signaling were investigated and validated for each strand. They find that

miR-34a-5p inhibited proliferation, migration and cell invasion accompanied by MMP9 activity and MAP2 protein reduction.

miR-34b-5p and miR-34c-5p inhibit proliferation and migration but not invasion. In contrast, miR-34c-5p inhibited MMP9 activity and MAP2 protein

miR-34b-5p has no effect over these genes

miR-34a-3p and miR-34b-3p inhibits proliferation and migration but not invasion despite the later reduces MMP2 activity

miR-34c-3p inhibited proliferation, migration and cell invasion accompanied by MMP9 activity and MAP2 protein inhibition.

The difference in cellular processes, MMP2, MMP9 activity and MAP2 protein inhibition by miR-34 family members suggests the participation of other regulated-genes. Hence, the study by Córdova-Rivas et al. provides insights into the roles of strand* of miR-34 family in cervical cancer

The experiments are well carried out and the manuscript is well written. Hence the paper highlights the complexity Hence, the study adds import knowledge to our view on the biology of circulating miRNAs and whether they are messengers or just passengers. As has been shown for hormones variation in the plasma concentration can be due to changes in the amount of binding proteins, while the amount biological active hormone is unaffected. Therefore, studies like the present are important as they give knowledge of how much of a messenger is found in the different compartments in the plasma. This is important for both evaluating changes in their concentration and for predicting biological function.

I only have a few minor comments to this well experimentally well performed and interesting paper

In silico analysis: These are very fine but have some limitations. Although some are very refined, they are in silico and not experimental. Therefore, the authors should in figure 3 indicate which of the targets have been experimentally validated and which not.

Secondly for a microRNA to be effective both the microRNA and its targets must be expressed in the same cell (that is excluding circulating miRNA). Therefore, the authors should in figure 3 indicate which of the targets are expressed in the cervical cancers.

Author Response

Dear reviewer 1, thank you for yours comments and suggestions we tried to address all issues in the Review Report and Comments and Suggestions.

Question and Reviewer answer.

Are the results clearly presented? Can be improved

Author’s improvement

Please check: 

Line 171: In Figure 2. “Figure 2. Ectopic expression of miR-34 family members affects cell migration and invasion in SiHa cells” the word invasion was added.

Line 180 and 187: We searched targets for MMP2, MMP9 and MAP2 in miRWalk database because reviewer 2 mentioned this database in comments and suggestions. Therefore we added the miRWalk database in the result section.

In line 206 we changed diminish for reduction.

In line 207 we corrected a typographic error, ausent was changed by absent. 

Additionally, we changed RNA22-HAS for RNA22-HSA and this modification is marked in red in the manuscript.

Comments and suggestions made by reviewer

I only have a few minor comments to this well experimentally well performed and interesting paper 

1.- In silico analysis: These are very fine but have some limitations. Although some are very refined, they are in silico and not experimental. Therefore, the authors should in figure 3 indicate which of the targets have been experimentally validated and which not. 

Author’s Response: in Figure 3 description we indicated validated targets as well as not validated as you kindly suggested.

Please check lines 258-260, 262-263, 266-277, 272-274, 276-277, and 279-280.

2.-Secondly for a microRNA to be effective both the microRNA and its targets must be expressed in the same cell (that is excluding circulating miRNA). Therefore, the authors should in figure 3 indicate which of the targets are expressed in the cervical cancers. 

Author’s Response: We added references to indicated which genes are expressed in cervical cancer as well as which are over-expressed in others cancers suggesting a participation in carcinogenesis. 

Please check lines 280 to 285.

Reviewer 2 Report

The strategy of miRNA target genes selection is not very appropriate. 

The authors used some bioinformatic tools and data bank for miR-34 target gene selection, but not in validated target. According to this, the results may be invalid.

in miRWalk, validated miRNA-Gene target, there is not MMP9 as target of miR-34.

Author Response

Dear reviewer 2, thank you for yours comments and suggestions we tried to address all issues in the Review Report and Comments and Suggestions.

Question and Reviewer answer.

Does the introduction provide sufficient background and include all relevant references? Must be improved

Author’s improvement

Please review: 

Line 50 and 51: We improved the introduction and we changed reference 1.

Line 115 to 118: We improved the introduction and we added references 45 to 48.

Question and Reviewer answer.

Is the research design appropriate? Must be improved

Author’s response:

Regarding the design of research we believe it is appropriate, however we are open to suggestions. 

Question and Reviewer answer.

Are the methods adequately described?  Can be improved

Author’s improvement

Lines 382-384 and 386-388: Transfection assay was corrected and more detailed.

Lines 397-398: Cell proliferation assay was more detailed.

Lines 403-404 and 407: Cell migration and invasion assays were more detailed.

Line 415 and 416: Zymography assay was more detailed.

Lines 431-435: Immunobloting assay was more detailed.

Question and Reviewer answer.

Are the results clearly presented? Yes

Author’s improvement

Line 171: In Figure 2. “Figure 2. Ectopic expression of miR-34 family members affects cell migration and invasion in SiHa cells” the word invasion was added.

Line 180 and 187: We searched targets for MMP2, MMP9 and MAP2 in miRWalk database because the reviewer mentioned this database in comments and suggestions. Therefore we added the miRWalk database in the result section.

Additionally, we changed RNA22-HAS for RNA22-HSA and this modification is marked in red in the manuscript.

Question and Reviewer answer.

Are the conclusions supported by the results? Can be improved.

Author’s response

About conclusions we think that they are well supported by the results presented in the paper.

Comments and suggestions make it by reviewer

1.-The strategy of miRNA target genes selection is not very appropriate. 

Author’s Response: We used several databases as well as validated targets, therefore we think target genes selection is appropriate.

Yang et al 2018 (International journal of oncology 201751, 378-388) demonstrated that miR-34a-5p targets MMP2 and MMP9 taking in consideration RNA22 database.

MMP2 is a validated target for miR-34c-3p in DIANA, and miRanda databases; for miR-34c-5p in DIANA and RNA22; for miR-34a-5p in DIANA database.

MAP2 is a validated target for miR-34b-3p and miR-34c-3p in DIANA, miRanda and PITA databases and for miR-34a-5p in DIANA database.

2.-The authors used some bioinformatic tools and data bank for miR-34 target gene selection, but not in validated target. According to this, the results may be invalid. 

Author’s Response: We searched in miRSystem that comprised 7 databases that show validated targets; therefore we think the results are valid.

Yang et al 2018 (International journal of oncology 201751, 378-388) demonstrated that miR-34a-5p targets MMP2 and MMP9 taking in consideration RNA22 database.

MMP2 is validated target for miR-34c-3p in DIANA, and miRanda databases; for miR-34c-5p in DIANA and RNA22; for miR-34a-5p in DIANA database.

MAP2 is validated target for miR-34b-3p and miR-34c-3p in DIANA, miRanda and PITA databases and for miR-34a-5p in DIANA database.

3.-in miRWalk, validated miRNA-Gene target, there is not MMP9 as target of miR- 34. 

Author’s Response: in miRWalk MMP9 appears as target of miR-34a-5p and miR-34c-5p. Yang et al 2018 (International journal of oncology 201751, 378-388) demonstrated that miR-34a-5p targets MMP9 taking in consideration RNA22 database. Therefore we considerate it as validated and it is specified this way in the manuscript.

.